# Performance Evaluation of Automated Erythrocyte Sedimentation Rate (ESR) Analyzers in a Multicentric Study

**DOI:** 10.3390/diagnostics14182011

**Published:** 2024-09-11

**Authors:** Flaminia Tomassetti, Cinzia Calabrese, Fabio Bertani, Michele Cennamo, Daniela Diamanti, Alfredo Giovannelli, Roberto Guerranti, Roberto Leoncini, Maria Lorubbio, Agostino Ognibene, Eleonora Nicolai, Martina Pelagalli, Carolina Pieroni, Sergio Bernardini, Massimo Pieri

**Affiliations:** 1Department of Experimental Medicine, University of Rome Tor Vergata, Via Montpellier 1, 00133 Rome, Italy; flaminia.tomassetti@students.uniroma2.eu (F.T.); cal.cinzia04@gmail.com (C.C.); alfredo.giovannelli@gmail.com (A.G.); nicolai@med.uniroma2.it (E.N.); pelagallimartina90@gmail.com (M.P.); bernards@uniroma2.it (S.B.); 2Department of Laboratory Medicine, Tor Vergata University Hospital, Viale Oxford 81, 00133 Rome, Italy; 3Clinical Pathology and Microbiology Unit, Laboratory Analysis, ASST Lariana, Hospital Sant’Anna, 22100 Como, Italy; fbertani26@gmail.com (F.B.); michelecennamo1@gmail.com (M.C.); 4Department of Translational Medical Sciences, University of Naples “Federico II”, 80126 Naples, Italy; 5Research and Development Department, DIESSE Diagnostica Senese S.p.A., Monteriggioni, 53035 Siena, Italy; danieladiamanti@diesse.it (D.D.); carolinapieroni@diesse.it (C.P.); 6Clinical Pathology Unit, Innovation, Experimentation and Clinical and Translational Research Department, University Hospital of S. Maria alle Scotte of Siena, Viale Mario Bracci 16, 53100 Siena, Italy; roberto.guerranti@unisi.it (R.G.); roberto.leoncini@unisi.it (R.L.); 7Chemical-Clinical Analysis Laboratory, Department of Laboratory Medicine and Transfusion, San Donato Hospital, 52100 Arezzo, Italy; maria.lorubbio@uslsudest.toscana.it (M.L.); agostino.ognibene@uslsudest.toscana.it (A.O.)

**Keywords:** ESR, multicentric study, inflammation, method validation

## Abstract

Background: Erythrocyte Sedimentation Rate (ESR) is an easy test used to diagnose and monitor inflammatory and infectious diseases. The aim of this study was the evaluation of the performance of three ESR automated analyzers, VES-MATIC 5, CUBE 30 TOUCH, and MINI-CUBE, involving four Italian polyclinics in Rome, Siena, Como, and Arezzo, as well as inter-site variability assessment to detect possible device-dependent and operator-dependent influences. Methods: Accuracy analysis was carried out by analyzing the same samples with all three instruments and comparing them with the Westergren method. Precision was assessed with quality control material through intra-run and inter-run precision. Repeatability was estimated by reanalyzing fresh blood samples belonging to three ESR ranges (low, intermediate, and high) six times. Results: The results showed a strong correlation (Spearman coefficients R^2^) between the manual method and VES-MATIC 5 (0.978), CUBE 30 TOUCH (0.981), and MINI-CUBE (0.974). The accuracy of all clinics was excellent, with coefficients of variation (CVs) of less than 10% for all instruments. Repeatability confirmed an excellent level for all ESR ranges, with CVs below 10%. Conclusions: The study proved that all three automated instruments offer optimal performance for accuracy and precision and are suitable for both large and small facilities without influences of the laboratory environment.

## 1. Introduction

Many pathologies (infections, autoimmune diseases, oncological conditions) are associated with inflammation, either acute or chronic. During the inflammatory process, the production of molecules known as acute phase proteins (APPs) increases [1]. The APPs are highly conserved plasma proteins that are increasingly secreted in response to various injuries, regardless of their location and cause, to support the systemic regulation of defense processes, such as coagulation, proteolysis, and tissue repair [2,3]. Various APPs have been applied as diagnostic parameters for a long time; for example, fibrinogen, C-reactive protein (CRP), serum amyloid A (SAA), metal-binding proteins, lysozyme, lectin, etc. Moreover, some of them interact directly with red blood cells (RBCs) and influence the physical phenomenon defined as the Erythrocyte Sedimentation Rate (ESR) [4], whose measure represents an evaluation of inflammatory status.

ESR is the velocity (expressed in mm/h) measuring how fast red blood cells (RBCs) sediment at the bottom of a test tube that is maintained in a vertical position within 1 h [5]. It is composed of three steps: the aggregation phase, with “rouleaux” formation [6]; the precipitation phase, in which an increase in cathodal proteins, such as immunoglobulins and fibrinogen, stabilizes the rouleaux formation [7]; and last, the precipitation phase [8]. In the packing phase, the red cell aggregates pile up on the bottom of the tube.

Although ESR is a sensitive indicator of inflammation, it generally lacks the specificity necessary to identify the exact cause of a disease [3]. Despite this inability, the test is widely utilized due to its ease and economy by clinicians in the diagnosis and monitoring of many diseases.

Moreover, higher levels of ESR are an important diagnostic criterion for rheumatology disease, likewise, for various types of cancers that have an overall poor prognosis, including Hodgkin’s neoplastic diseases and multiple myeloma [9].

The reference method for measuring ESR, proposed by the International Committee for Standardization in Haematology (ICSH), is based on the findings described by Westergren a century ago [10]. The Westergren method measures the distance (in millimeters) that RBCs in citrate anticoagulated whole blood (ratio 4:1) fall to the bottom of a standardized, upright, elongated tube over 1 h due to the influence of gravity. These tubes are made of either glass or plastic, with an internal diameter of 2.55 +/− 0.15 mm and lengths of 200 mm [8].

Nowadays, continuous technical innovations have been adopted, and several new automated or semi-automated methods have become available. These innovations and the use of closed blood collection tubes have significantly improved existing procedures, delivering superior performance by reducing biohazard risks to operators, decreasing the time required to perform ESR, and guaranteeing results traceable to the reference method.

The ICSH established a Working Group to investigate these new approaches and compile recommendations for their validation and verification [11]. The validations of the ESR automated analyzers the MINI-CUBE, CUBE 30 TOUCH, and VES-MATIC 5 (DIESSE Diagnostica Senese S.p.A., Monteriggioni, Italy), according to the ICSH guidelines, have already been published [11,12,13,14,15,16], as well as the method comparison between analyzers employing different technologies [12,16,17,18], but a multi-site investigation of these analyzers has not yet been conducted.

The purpose of this study was the performance evaluation of the three ESR automated devices in terms of accuracy concerning the gold standard Westergren method and inter-laboratory precision in a multicentric study. Regardless of the instrument and site where the patient is monitored, the same reliability must be guaranteed. The final goal was the investigation of inter-instrument performance and inter-site variability, due respectively to possible device-dependent and operator-dependent influences from blood collection to sample management and analysis, by addressing the recommendations of the International Council for Standardization in Haematology [15].

## 2. Materials and Methods

### 2.1. Study Design

The investigation was conducted as a multicenter study involving four Italian clinical centers: Tor Vergata University Polyclinic in Rome, Santa Maria alle Scotte University Hospital in Siena, Sant’Anna Hospital in Como, and San Donato Hospital in Arezzo. All experimental activities were conducted in the same period, from November 2023 to May 2024.

The proposed project was divided into 3 phases: Accuracy assessment, Precision study, and Repeatability study.

Phase I: For the Accuracy assessment, each site consecutively analyzed the same fresh blood samples with all three instruments and, finally, used the Westergren manual method within 4 h of sampling to ensure stability. Samples were selected to cover the three analytical ranges of ESR (low, medium, high within instrumental range of 1–140 mm/h).

Phase II: The Precision study was characterized by the intra-run, inter-run, and total precision evaluated for all three instruments with the same lot of ESR control cube for every center (DIESSE Diagnostica Senese S.p.A. Monteriggioni, Siena, Italy). The quality control (QC) material was sent by manufacturing to all laboratories and included a normal (low ESR) and abnormal level (high ESR) [19]. Each ESR control was analyzed for 5 consecutive days, twice a day (once in the morning and once in the afternoon) by the same operator [20].

Phase III: The repeatability evaluation was carried out by selecting fresh blood samples for each analytical range of ESR (low, medium, high within the instrumental range of 1–140 mm/h), which was repeated 6 times [21]. The samples must necessarily be different for each instrument to avoid exceeding sample stability.

### 2.2. Specimen Collection

For each center, the EDTA blood leftover samples were randomly selected from the daily routine. All blood samples were leftovers from the daily blood samples, including both ambulatory and hospitalized patients. Samples were anonymized and analyzed after all the other tests requested were completed. For accuracy analysis, Rome Tor Vergata Hospital collected 230 blood samples. Siena polyclinic collected 191 blood samples. The number of samples collected at hospitals in Como and Arezzo were 192 and 174, respectively. For repeatability analysis, the samples collected from every single center involved the following ranges: low ESR values (<20 mm/h), intermediate ESR (21–60 mm/h), and high ESR (>60 mm/h).

All blood samples were analyzed within 4 h of collection to ensure stability in the EDTA tubes [22].

### 2.3. ESR Analyzers Description

The three analyzers (VES-MATIC 5, CUBE 30 TOUCH, and MINI-CUBE, from DIESSE Diagnostica Senese S.p.A. Monteriggioni, Siena, Italy) are very similar in terms of methods for ESR measure and software but differ in their maximum simultaneous loading capacity, load mode, and additional features. They are classified as modified Westergren methods [11] because they are based on reference one, but they have a shorter assay time, and the blood is diluted with EDTA as an anticoagulant and not with citrate. The ESR calculation is carried out by measuring the plasma column obtained after 20 min of sedimentation covering all three phases of erythrocyte sedimentation: aggregation, sedimentation, and packing. The sedimentation is revealed by the optoelectronic light group, and a special algorithm is subsequently applied to translate the obtained value into Westergren data and corrected for temperature variations, according to Manley correction [23]. In each instrument, QCs can be processed before each analytical measurement and are automatically stored by the instrument to generate the Levey-Jannings control chart. All devices operate without the need for reagents or waste bids, preventing operators’ exposure to biological materials; therefore, there are no ordinary and extraordinary maintenance or supplemental costs.

#### 2.3.1. VES-MATIC 5

This new automatic instrument is equipped with an automatic stirring system and an internal barcode reader [15,16]. Briefly, all samples are collected in an EDTA anticoagulated tube and loaded into the instrument directly on the same rack as the complete blood count (CBC) instruments. Samples are mixed and under controlled temperature evaluated for sedimentation measurement using an optoelectronic light source. The sedimentation rate is evaluated by multiple optical recordings for 20 min, and the differences are estimated. The first result is available after 28 min. Technologically innovative, the new analyzer applies an AI system in recognition of lipemic, hemolyzed, coagulated, or mislabeled samples, while the Internet of Things, through an internal camera, is used for advanced remote assistance. In this way, the analyzer is connected directly to diagnostic devices, reporting any malfunction directly and receiving instructions for self-repair, where possible. The throughput is 190 samples per hour, and walk-away mode and continuous loading are supported.

#### 2.3.2. CUBE 30 TOUCH

CUBE 30 TOUCH, like VES-MATIC 5, is equipped with an automatic stirring system and an internal barcode reader. Unlike the previously mentioned model, loading is manual, allowing a maximum of 30 samples to be analyzed at a time and up to 90 samples per hour. The instrument performs direct testing from EDTA tubes without any handling or consumption of patient samples. Results are processed within 20 min.

#### 2.3.3. MINI-CUBE

The analyzer can measure ESR directly from the capped EDTA tube with four simultaneous and random-access samples for an executive period of 20 min. The samples, before the analysis, are manually gently mixed by turning them upside down and put inside the instrument, which allows the continuous loading of samples for up to four analyses simultaneously and throughput of 12 samples/hour. The barcode reader is external.

#### 2.3.4. Westergren Method

A total of 1.0 mL of venous blood (K3EDTA) was collected manually via pipettes into a tube containing 0.250 mL of sodium citrate. After being mixed gently for a minimum of eight times, blood was drawn into the standardized Westergren tube, marked to 200 mm. This tube was then placed vertically in a rack at room temperature for 1 h. The visual determination of the result was identified by the mark where the upper limit of erythrocyte sedimentation had settled due to gravity. The Westergren method was performed manually, according to the ICSH’s recommendations, within 4 h of blood sampling [24,25].

### 2.4. Statistics

Passing–Bablok correlation analysis was evaluated to compare the instruments to the Westergren test, using the nonparametric measure of rank correlation (statistical dependence between the rankings of two variables) and the Spearman rank correlation coefficient (R^2^).

Bland–Altman test was performed as an alternative analysis to determine the agreement between the different instruments concerning the gold standard (Westergren) and to investigate any possible relationship of the discrepancies between the measurements and the true value.

The total sample size for accuracy investigations was estimated by the sample size calculator in “CREATIVE RESEARCH SYSTEM” (https://www.surveysystem.com/sscalc.htm accessed on 8 August 2024) at a confidence level of 95%, confidence interval of 0.5, population of 800 (*n* 200 × 4 sites). The sample size determined by the algorithm was 784 enrolled patients. Samples should be equally balanced.

The coefficients of variation (CVs) were calculated as the ratio between the standard deviation and the mean value and expressed as %. The CVs obtained from the study were compared with the ones declared by the manufacturer.

All data were examined using Med Calc Ver.18.2.18 (MedCalc Software Ltd., Ostend, Belgium).

The statistical significance level established for all tests performed was *p* < 0.05.

## 3. Results

Each hospital center analyzed blood samples in EDTA tubes to evaluate the performance of the three instruments (VES-MATIC 5, CUBE 30 TOUCH, and MINI-CUBE) compared with the reference method (Westergren).

### 3.1. Correlation Analysis

In Table 1, all the Passing–Bablok regressions and the Bland–Altman analyses for each center and each automated analyzer are presented separately.

Moreover, a cumulative analysis of all 787 data from the four hospital centers was carried out, as shown in Figure 1. There was a good correlation with a Spearman coefficient (R^2^) of 0.981 (Confidence Interval, CI 95%: 0.979 to 0.984) between the CUBE 30 TOUCH and the Westergren method, an R^2^ of 0.977 (CI 95%: 0.974 to 0.980; *p* < 0.0001) between the MINI-CUBE and the Westergren method, and R^2^ of 0.978 (CI 95%: 0.975 to 0.98; *p* < 0.0001) was reached between VES-MATIC 5 and the Westergren method.

### 3.2. Precision Study

The precision study was evaluated using measurements at each quality control (QC) level, high and low, for a total of 5 days and three times per day, following the Clinical and Laboratory Standards Institute (CLSI) Protocol EP15-A3 [26]. Table 2 shows the coefficients of variation (CVs) for the instrumental precision of each hospital center. The results for the four centers, for all instruments, and the two QC levels highlighted excellent precision, with a CV < 10%. It should be noted that the coefficients of variation at the lower level had the highest CVs due to the small numbers measured.

Furthermore, a cumulative analysis of all 787 data from the four hospital centers was performed to establish the overall precision. In Table 3, the CVs for the total instrumental precision of all the hospitals are reported. The overall CVs were in line with those declared by the manufacturer, and the results met the accepting criteria (Table 3).

### 3.3. Repeatability

Repeatability was estimated using patient samples collected from the laboratory routine and not included in the correlation analysis, with ESR values belonging to each interval and analyzed six times. The intervals considered were the following: samples with low ESR (<20 mm/h), intermediate ESR (21–60 mm/h), and high ESR (>60 mm/h). Table 4 shows the CV for the instrumental repeatability of each hospital center. The results showed that for all centers and all instruments, an excellent repeatability level of less than 10% (CV < 10%).

## 4. Discussion

Erythrocyte Sedimentation Rate (ESR) is a non-specific disease index, which is not diagnostic for any disease. However, an elevated ESR may indicate the presence of inflammation, infection, rheumatological disease, or cancer. ESR tests, in combination with patients’ clinical history and physical examination, aid in the diagnosis, management, and follow-up of different autoimmune diseases, acute and chronic infections, and tumors. Moreover, some significant and frequent alterations of ESR are common in anemic patients, where ESR could be falsely higher due to the low RBC count [27].

In a global world where people are constantly moving, it is necessary for the patient that diagnostic instruments offer the same accuracy and reliability. The implementation of automated ESR analyzers resolves almost completely the second issue. The remaining questions are to verify whether different laboratories using the same Good Practice Laboratories (GPL) and the same model of ESR instruments work with the same accuracy.

Some other studies have already assessed the good performance and the high precision of the ESR analyzers [11,12,13,14,15,16,28], while others have compared DIESSE technologies with other methods [16,17,18]. However, no one has assessed their performance in a multicentric study, except for a work about the ESR analyzer Test-1 based on an alternative method (Alifax, Padova Italia) that was conducted in three different hospitals [29]. Hence, this is the first inter-laboratory study performed on modified Westergren ESR analyzers from the same company that works under the same conditions and protocols. Moreover, the strengths of a multicentric study guarantee diverse population coverage and uniformity of analytical methods to minimize changes in data fishing and inappropriate analysis [30].

Regarding accuracy evaluation, a strong correlation was observed from the data obtained from the cumulative statistical analysis (*n* = 787) between each instrument and the Westergren method, with a Spearman coefficient (R^2^) of 0.978 between VES-MATIC 5 and the Westergren method, 0.981 between CUBE 30 TOUCH and Westergren method, and 0.976 between MINI-CUBE and Westergren method. The average biases obtained can be considered acceptable from a clinical point of view. Moreover, the results from a single center of method comparison showed a similar excellent correlation. No differences came out from the elaboration of pooled data concerning the single-center data. Overall, these results confirm the optimal correlations observed for each device investigated individually in previous studies [14,15]. These findings support the good accuracy and reliability of the instrument without influences of the laboratory environment, ensuring that patients’ results are consistent in different healthcare facilities. Moreover, in the case of the presence of different models of ESR analyzers, the data guarantee that patients can obtain similar ESR results regardless of the instrument used in the same laboratory, ensuring a reliable diagnosis and monitoring of treatment.

ESR instruments do not require internal calibrations before use, although, as with any other clinical laboratory test, good laboratory practices suggest analyzing ESR for quality control (QC) and monitoring the CV% every day before routine practice [22]. Therefore, precision analysis was conducted over 5 days with high and low ESR controls and showed excellent intra-run and inter-run precision, with CVs consistently below 10% across all centers and instruments (Table 1, Table 2, Table 3 and Table 4). Still, higher CVs were observed at very low ESR levels, where minor result variations can lead to increased standard deviations. These findings align with previous studies and confirm that the precision levels achieved are consistent with manufacturer specifications [14,31,32]. Furthermore, the coefficients of variation for the total instrumental precision of all clinics were optimal and in line with the declared cut-off. Additionally, using the same lot of ESR QC across all centers effectively simulates an External Quality Assessment (EQA) program [33]. The results guaranteed high-quality analytical performances in the ESR test are reliable and valuable for inter-laboratory proficiency testing [24,34].

Repeatability was estimated using patient samples not included in the correlation analysis, with ESR values belonging to each range and analyzed six times. The intervals considered were the following: samples with low ESR (<20 mm/h), intermediate ESR (21–60 mm/h), and high ESR (>60 mm/h). The results showed, for all centers and for all instruments, an excellent repeatability level of less than 10% imprecision (CV < 10%). This result guarantees that in the rare case of test repetition, patients receive the same results (provided that the test is carried out inside the sample’s stability window) and demonstrates the sensitivity of the instruments, allowing reliable monitoring of ESR values in patients. In conclusion, our study demonstrates the optimal performance of all the instruments, producing valid results in terms of precision, accuracy, and comparison, even if carried out in different hospital facilities. These results highlight the trustworthiness of the three analyzers that can easily adapt to the needs of any clinical laboratory, offering the same quality results, even if they have different loading capacities. Furthermore, since all the instruments are maintenance-free and have no sample manipulation, blood sample consumption, or production of potentially infectious liquid waste, they are valid methods for measuring ESR in the routine of both large and small medical laboratories.

A limitation of this study is the accuracy evaluation since it was not possible to use the same patient samples in different hospitals given the distance between the centers and the impossibility of carrying out the test within 36 h of collection, unlike what was performed by Cha et al. [29]. Their experimental design could be the cause of the differences between one hospital and the others, as declared by the same authors.

## 5. Conclusions

Finally, this study proved that all hospitals involved offered a standardized workflow, employing good laboratory practices from sample collection, handling, and analysis to final reports. Our work has demonstrated that automated ESR analyzers (VES-MATIC 5, CUBE 30 TOUCH, and MINI-CUBE) deliver consistent performance in terms of precision, accuracy, and reliability across various clinical settings. Undoubtedly, the good results in terms of instrument and laboratory performance are an indirect consequence of the guidelines proposed by the ICSH over the years to standardize the ESR analyzers and laboratory protocols [11,35].

Besides the optimal results, these instruments offer significant advantages in practicality, speed, cost-efficiency, and accessibility. Specifically, these devices are more practical, as they eliminate the need for reagents, require no maintenance, and minimize biohazard risks by using closed systems, reducing operational costs. Moreover, they provide rapid results within 20 to 28 min, significantly improving turnaround time compared with the 1-h Westergren method (1 h). Furthermore, their compact design and user-friendly interfaces make them easy to integrate into both large and small laboratories.

These findings recommend the broader adoption of these analyzers, ensuring uniform diagnostic standards across various healthcare systems and, importantly, this multicentric approach underlines the critical role that such studies play in validating medical devices across diverse clinical environments and leads to standardization.

## Figures and Tables

**Figure 1 diagnostics-14-02011-f001:**
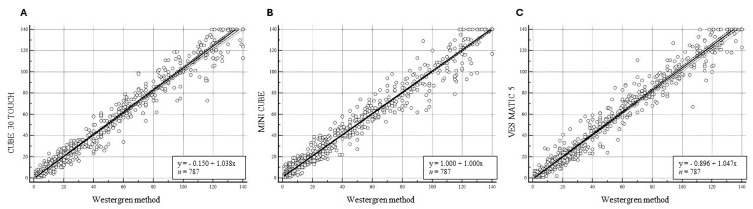
Passing–Bablok analysis of the three ESR analyzers for the overall samples. (**A**) Correlation between CUBE 30 TOUCH and Westergren Method, regression line: y = −0.1506 + 1.037x. (**B**) Correlation between MINI-CUBE and Westergren Method, regression line: y = 1.000 + 1.000x. (**C**) Correlation between VES-MATIC 5 and Westergren Method, regression line: y = −0.896 + 1.047x.

**Table 1 diagnostics-14-02011-t001:** Passing–Bablok regression and Bland–Altman analysis for the three ESR analyzers for the four hospitals.

		CUBE 30 TOUCH vs. Westergren Method	MINI-CUBE vs. Westergren Method	VES-MATIC 5 vs. Westergren Method
Tor Vergata Hospital	Sample size (*n*)	230	230	230
	Regression equation	y = −0.076 + 1.039x	y = 2.000 + 1.000x	y = −0.816 + 1.072x
	Spearman correlation coefficient (R^2^) and [95% CI]	0.970 [0.961 to 0.977]	0.958 [0.946 to 0.968]	0.972 [0.964 to 0.979]
	Significance level	*p* < 0.0001	*p* < 0.0001	*p* < 0.0001
	Bland–Altman mean [min.–max]	0.5 [min. −15.9; max −14.8]	−2.1 [min. −16.2; max 12.1]	−2.0 [min. −15.3; max 11.2]
Arezzo Hospital	Sample size (*n*)	174	174	174
	Regression equation	y = −0.104 + 1.042x	y = 3.000 + 1.000x	y = −1.353 + 1.068x
	Spearman correlation coefficient (R^2^) and [95% CI]	0.976 [0.967 to 0.982]	0.971 [0.961 to 0.978]	0.974 [0.965 to 0.981]
	Significance level	*p* < 0.0001	*p* < 0.0001	*p* < 0.0001
	Bland–Altman mean [min.–max]	1.5 [min. −10.2; max 13.2]	2.5 [min. −10.5; max 15.6]	0.4 [min. −13.0; max 13.7]
Siena Hospital	Sample size (*n*)	191	191	191
	Regression equation	y = −0.055 + 1.027x	y = 1.000x	y = −0.104 + 1.001x
	Spearman correlation coefficient (R^2^) and [95% CI]	0.989 [0.985 to 0.991]	0.988 [0.984 to 0.991]	0.988 [0.984 to 0.991]
	Significance level	*p* < 0.0001	*p* < 0.0001	*p* < 0.0001
	Bland–Altman mean [min.–max]	1.9 [min. −11.4; max 15.1]	2.5 [min. −10.5; max 15.6]	−0.1 [min. −15.2; max 14.9]
Como Hospital	Sample size (*n*)	192	192	192
	Regression equation	y = −0.903 + 1.041x	y = 0.472 + 0.973x	y = 0.125 + 1.050x
	Spearman correlation coefficient (R^2^) and [95% CI]	0.986 [0.981 to 0.989]	0.980 [0.973 to 0.985]	0.965 [0.954 to 0.973]
	Significance level	*p* < 0.0001	*p* < 0.0001	*p* < 0.0001
	Bland–Altman mean [min.–max]	1.9 [min. −11.4; max 15.1]	0.2 [min. −12.8; max 13.2]	2.3 [min. −12.8; max 17.3]

**Table 2 diagnostics-14-02011-t002:** Precision within run, between run, and total precision of each ESR analyzer for the four hospitals.

Tor Vergata Hospital				
	QC Level	CV % Within Run	CV % Between Run	Total Precision
VES-MATIC 5	High	2.8	2.8	3.6
VES-MATIC 5	Low	6.5	7.7	9.4
CUBE 30 TOUCH	High	3.3	2.9	3.9
CUBE 30 TOUCH	Low	9.1	4.7	9.0
MINI-CUBE	High	3.2	0.8	2.7
MINI-CUBE	Low	9.1	4.4	8.5
Arezzo Hospital				
	QC Level	CV% Within Run	CV% Between Run	Total Precision
VES-MATIC 5	High	2.7	0.7	2.3
VES-MATIC 5	Low	8.9	7.1	10.6
CUBE 30 TOUCH	High	4.2	2.4	4.2
CUBE 30 TOUCH	Low	10.3	4.3	9.7
MINI-CUBE	High	2.3	2.1	2.8
MINI-CUBE	Low	9.1	7.1	10.0
Siena Hospital				
	QC Level	CV % Within Run	CV % Between Run	Total Precision
VES-MATIC 5	High	2.0	1.4	2.2
VES-MATIC 5	Low	8.6	4.9	8.4
CUBE 30 TOUCH	High	1.7	1.4	2.0
CUBE 30 TOUCH	Low	8.6	5.1	8.8
MINI-CUBE	High	1.8	0.8	1.7
MINI-CUBE	Low	6.5	3.8	6.6
Como Hospital				
	QC Level	CV % Within Run	CV % Between Run	Total Precision
VES-MATIC 5	High	3.0	4.1	4.7
VES-MATIC 5	Low	11.1	3.1	9.4
CUBE 30 TOUCH	High	1.7	1.0	1.7
CUBE 30 TOUCH	Low	9.1	4.7	9.0
MINI-CUBE	High	1.5	2.3	2.6
MINI-CUBE	Low	9.1	4.7	9.0

**Table 3 diagnostics-14-02011-t003:** The overall precision of the three automated ESR analyzers.

	CUBE 30 TOUCH		VES-MATIC 5		MINI-CUBE	
	DIESSE	Evaluated	DIESSE	Evaluated	DIESSE	Evaluated
QC High	3.5	3.70	7.7	4.90	14.1	4.90
QC Low	9.8	1.30	13.3	2.02	8.3	1.50
	Accepted	Accepted	Accepted	Accepted	Accepted	Accepted

**Table 4 diagnostics-14-02011-t004:** Repeatability analysis of each ESR analyzer for the four hospitals.

Tor Vergata Hospital					
	Analytic Interval	*n*	MEAN	SD	CV (%)
VES-MATIC 5	Low	68	7.64	0.49	6
	Middle	31	29.57	2.9	9
	High	54	102.68	4.99	6
CUBE 30 TOUCH	Low	76	7.39	0.56	8
	Middle	51	1.74	1.74	6
	High	39	3.04	3.04	3
MINI-CUBE	Low	36	10.58	0.80	6
	Middle	22	39.41	2.96	6
	Up	30	100.45	3.22	4
Arezzo Hospital					
	Analytic Interval	*n*	MEAN	SD	CV (%)
VES-MATIC 5	Low	66	9.25	0.49	5
	Middle	32	29.01	2.22	8
	High	43	94.07	2.39	3
CUBE 30 TOUCH	Low	44	8.14	0.55	7
	Middle	42	30.36	1.16	4
	High	43	92.12	1.88	2
MINI-CUBE	Low	36	11.14	0.54	5
	Middle	25	32.63	1.19	4
	High	50	94.19	1.87	2
Siena Hospital					
	Analytic Interval	*n*	MEAN	SD	CV (%)
VES-MATIC 5	Low	75	7.80	0.56	7
	Middle	34	31.60	1.3	4
	High	53	91.75	2.25	2
CUBE 30 TOUCH	Low	81	8.0	0.5	6
	Middle	49	30.37	1.18	4
	High	40	88.41	2.00	2
MINI-CUBE	Low	37	10.75	0.51	5
	Middle	27	37.15	0.98	3
	High	32	95.14	1.67	2
Como Hospital					
	Analytic Interval	*n*	MEAN	SD	CV (%)
VES-MATIC 5	Low	17	8.75	0.82	9.4
	Middle	16	25.69	2.30	8.9
	High	79	86.02	6.13	7.9
CUBE 30 TOUCH	Low	34	10.63	0.74	7
	Middle	12	29.14	2.80	9.9
	High	20	88.62	3.22	4.3
MINI-CUBE	Low	37	10.64	0.61	5.7
	Middle	46	37.34	2.97	8.6
	High	30	77.49	3.02	4.2

## Data Availability

Data are available on specific requests.

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
