# Peer review of "Performance Evaluation of Automated Erythrocyte Sedimentation Rate (ESR) Analyzers in a Multicentric Study"

_diagnostics, 2024, doi:10.3390/diagnostics14182011_

Round 1

Reviewer 1 Report

Comments and Suggestions for Authors

The Authors present a paper:" Performance-Evaluation of automated Erythrocyte sedimentation rate (ESR) in a multi centric study" interesting from a clinical point of view and innovative.

The paper is exhaustive but it needs some specific remarks:

1. ABSTRACT is fine

2. INTRODUCTION: is too long and detailed in the first 3 paragraph. It could be shortened (too scholastic)

3. MATERIAL and METHODS: there are no mentions about the costs of different procedures: Are all equal? Can you find  them easily?

4. RESULTS: well reported as well as the tables

5. DISCUSSION:  The first 2 paragraph report the same concepts of Introduction (redundant). In the list of causes of ESR alterations don't mention ANEMIA as possible augmented ESR responsible. They should more emphasize the important differences among the different procedures.

CONCLUSIONS: not present but it should be added. In the Conclusions the Authors have to indicate some recommendations for using these alternative devices instead of conventional one: a) practicality? b) short time requested?c) costs? d) easy to find? Based on this worthy experience they could give some regulations for readers.

Comments on the Quality of English Language

The language is fine but could be reviewed by a mother language translator

Author Response

We would like to thank the referees and editors for evaluating our manuscript.

We have modified our work and tried to address all the reviewers’ concerns and believe that our paper has improved its value. The revisions were highlighted in yellow.

Below a step-by-step response.

Reviewer 1:

The Authors present a paper:" Performance-Evaluation of automated Erythrocyte sedimentation rate (ESR) in a multi centric study" interesting from a clinical point of view and innovative.

The paper is exhaustive but it needs some specific remarks:

  1. ABSTRACT is fine

R: Thanks.

  1. INTRODUCTION: is too long and detailed in the first 3 paragraph. It could be shortened (too scholastic)

R: We agree with this comment; therefore, we deleted some parts.

  1. MATERIAL and METHODS: there are no mentions about the costs of different procedures: Are all equal? Can you find them easily?

R: We thank the Reviewer for pointing this out, and we added a sentence (lines 142-145) to emphasize this aspect.

  1. RESULTS: well reported as well as the tables

R: Thanks again for your kind comment.

  1. DISCUSSION:  The first 2 paragraph report the same concepts of Introduction (redundant). In the list of causes of ESR alterations don't mention ANEMIA as possible augmented ESR responsible. They should more emphasize the important differences among the different procedures.

R: Thanks, we accordingly summarized some concepts in the Discussion. Moreover, we included in the text the Anaemia, as ESR alterations, as well as we enhanced the differences between different procedures, as suggested.

CONCLUSIONS: not present but it should be added. In the Conclusions the Authors have to indicate some recommendations for using these alternative devices instead of conventional one: a) practicality? b) short time requested?c) costs? d) easy to find? Based on this worthy experience they could give some regulations for readers.

R: As advised, we added the Conclusion paragraph at the end of the text, addressing and summarizing all the findings and the remain questions.

Reviewer 2 Report

Comments and Suggestions for Authors

    Dear Sir

I have read the manuscript by Tommasetti F and collaborators:

Performance - Evaluation of Automated Erythrocyte Sedimentation Rate (ESR) Analyzers In A Multicentric Study

This is a comparison work in the determination of ESR with three commercial methods: VES MATI 5, CUBE 30TOUCH and MINI CUBE evaluated against the Westergren method. The interesting aspect is the multicenter design of the study.

Introduction

The section appears verbose and in my opinion it would be advisable to make it more concise by eliminating some sentences that are not essential to the presentation of the study. It is therefore recommended to eliminate the following sentences

lines 49-54

lines69-71

lines 77-88

lines 95-97

From lines 101 to 106 the characteristics of the analyzers used in the study are briefly described. These will be better described in the Materials and Methods section and therefore this part of the text can be eliminated.

Material and Methods

This section is rather long-winded but it is functional for an accurate description of the study design, the samples processed, the analysers used, the statistical methodologies used in the processing of the results obtained.

Results

This section appears well organized and usefully supported by graphs and tables. The results are not duplicated in the text.

Discussion

this section seems very wordy and, in my opinion, should be drastically shortened by eliminating some portions of text that do not add much to the discussion of the results obtained. It is therefore recommended to eliminate lines

lines 274 - 280

lines 284 - 293

The paragraph between line 300 and line 313 should be rewritten and made more concise and understandable.

The paragraph between line 333 and line 357 left me perplexed. Considering that NCCLS has established that the methods for determining the ESR do not require calibration I do not understand the rationale of the paragraph. The entire paragraph should be reworded in a more concise manner and justifying its rationale.

lines 376-380 should be eliminated.

The entire concluding paragraph (lines 386-397) should be rewritten making it more concise and understandable.

Comments on the Quality of English Language

English language need minor revision

Author Response

We would like to thank the referees and editors for evaluating our manuscript.

We have modified our work and tried to address all the reviewers’ concerns and believe that our paper has improved its value. The revisions were highlighted in yellow.

Below a step-by-step response.

Reviewer 2:

I have read the manuscript by Tomassetti F and collaborators:

Performance - Evaluation of Automated Erythrocyte Sedimentation Rate (ESR) Analyzers In A Multicentric Study

This is a comparison work in the determination of ESR with three commercial methods: VES MATI 5, CUBE 30TOUCH and MINI CUBE evaluated against the Westergren method. The interesting aspect is the multicenter design of the study.

Introduction

The section appears verbose and in my opinion it would be advisable to make it more concise by eliminating some sentences that are not essential to the presentation of the study. It is therefore recommended to eliminate the following sentences

lines 49-54

lines69-71

lines 77-88

lines 95-97

From lines 101 to 106 the characteristics of the analyzers used in the study are briefly described. These will be better described in the Materials and Methods section and therefore this part of the text can be eliminated.

 R: We agree with this comment; therefore, we deleted the lines, as suggested, and we summarized the sentence between lines 101 to 106 (now lines 80-84).

Material and Methods

This section is rather long-winded but it is functional for an accurate description of the study design, the samples processed, the analysers used, the statistical methodologies used in the processing of the results obtained.

R: Thanks.

Results

This section appears well organized and usefully supported by graphs and tables. The results are not duplicated in the text.

R: Thanks for your kind comment.

Discussion

this section seems very wordy and, in my opinion, should be drastically shortened by eliminating some portions of text that do not add much to the discussion of the results obtained. It is therefore recommended to eliminate lines

lines 274 – 280

R: Thanks, we deleted the sentence.

lines 284 – 293

R: We thank the reviewer for this suggestion; however, we think that these sentences highlight the aim of our study and present its novelty. Therefore, we limited to shorten up the text to a few lines (264-267).

The paragraph between line 300 and line 313 should be rewritten and made more concise and understandable.

R: We agree with the reviewer, and we modified the paragraph.

The paragraph between line 333 and line 357 left me perplexed. Considering that NCCLS has established that the methods for determining the ESR do not require calibration I do not understand the rationale of the paragraph. The entire paragraph should be reworded in a more concise manner and justifying its rationale.

R: We thank the Reviewer, therefore we formulated the concept in a different way (lines 293-306).

lines 376-380 should be eliminated.

R: Thanks, we deleted the sentence.

The entire concluding paragraph (lines 386-397) should be rewritten making it more concise and understandable.

R: Thank you for pointing this out. Hence, we modified the concluding sentences, and we included a “Conclusion” paragraph.

Round 2

Reviewer 1 Report

Comments and Suggestions for Authors

The revised version by the Authors is fine to me. The Authors followed with accuracy the suggestions received by reviewers.